# Revisiting Over-smoothing in BERT from the Perspective of Graph

**Han Shi**[1]*, **Jiahui Gao**[2]*, **Hang Xu**[3], **Xiaodan Liang**[4],
**Zhenguo Li**[3], **Lingpeng Kong**[2], **Stephen M.S. Lee**[2], **James T. Kwok**[1]
[1]Hong Kong University of Science and Technology, [2]The University of Hong Kong,
[3]Huawei Noah's Ark Lab, [4]Sun Yat-sen University
`{hshiac,jamesk}@cse.ust.hk,{sumiler,smslee}@hku.hk,lpk@cs.hku.hk,`
`{xu.hang,li.zhenguo}@huawei.com,xdliang328@gmail.com`

## Abstract

Recently over-smoothing phenomenon of Transformer-based models is observed in both vision and language fields. However, no existing work has delved deeper to further investigate the main cause of this phenomenon. In this work, we make the attempt to analyze the over-smoothing problem from the perspective of graph, where such problem was first discovered and explored. Intuitively, the self-attention matrix can be seen as a normalized adjacent matrix of a corresponding graph. Based on the above connection, we provide some theoretical analysis and find that layer normalization plays a key role in the over-smoothing issue of Transformer-based models. Specifically, if the standard deviation of layer normalization is sufficiently large, the output of Transformer stacks will converge to a specific low-rank subspace and result in over-smoothing. To alleviate the over-smoothing problem, we consider hierarchical fusion strategies, which combine the representations from different layers adaptively to make the output more diverse. Extensive experiment results on various data sets illustrate the effect of our fusion method.

## 1 Introduction

Over the past few years, Transformer (Vaswani et al., 2017) has been widely used in various natural language processing (NLP) tasks, including text classification (Wang et al., 2018a), text translation (Ott et al., 2018), question answering (Rajpurkar et al., 2016; 2018) and text generation (Brown et al., 2020). The recent application of Transformer in computer vision (CV) field also demonstrate the potential capacity of Transformer architecture. For instance, Transformer variants have been successfully used for image classification (Dosovitskiy et al., 2021), object detection (Carion et al., 2020) and semantic segmentation (Strudel et al., 2021). Three fundamental descendants from Transformer include BERT (Devlin et al., 2019), RoBERTa (Liu et al., 2019) and ALBERT (Lan et al., 2020), which achieve state-of-the-art performance on a wide range of NLP tasks.

Recently, Dong et al. (2021) observes the "token uniformity" problem, which reduces the capacity of Transformer-based architectures by making all token representations identical. They claim that pure self-attention (SAN) modules cause token uniformity, but they do not discuss whether the token uniformity problem still exists in Transformer blocks. On the other hand, Gong et al. (2021) observe the "over-smoothing" problem for ViT (Dosovitskiy et al., 2021), in that different input patches are mapped to a similar latent representation. To prevent loss of information, they introduce additional loss functions to encourage diversity and successfully improve model performance by suppressing over-smoothing. Moreover, "overthinking" phenomenon, indicating that shallow representations are better than deep representations, also be observed in (Zhou et al., 2020; Kaya et al., 2019). As discussed in Section 3, this phenomenon has some inherent connection with over-smoothing. In this paper, we use "over-smoothing" to unify the above issues, and refer this as the phenomenon that the model performance is deteriorated because different inputs are mapped to a similar representation.

As the over-smoothing problem is first studied in the graph neural network (GNN) literature (Li et al., 2018; Xu et al., 2018; Zhao & Akoglu, 2020), in this paper, we attempt to explore the cause of such

---

*Equal contribution.

problem by building a relationship between Transformer blocks and graphs. Specifically, we consider the self-attention matrix as the normalized adjacency matrix of a weighted graph, whose nodes are the tokens in a sentence. Furthermore, we consider the inherent connection between BERT and graph convolutional networks (Kipf & Welling, 2017). Inspired by the over-smoothing problem in GNN, we study over-smoothing in BERT from a theoretical view via matrix projection. As opposed to Dong et al. (2021), where the authors claim that layer normalization is irrelevant to over-smoothing, we find that layer normalization (Ba et al., 2016) plays an important role in over-smoothing. Specifically, we theoretically prove that, if the standard deviation in layer normalization is sufficiently large, the outputs of the Transformer stacks will converge to a low-rank subspace, resulting in over-smoothing. Empirically, we verify that the conditions hold for a certain number of samples for a pre-trained and fine-tuned BERT model (Devlin et al., 2019), which is consistent with our above observations.

To alleviate the over-smoothing problem, we propose a hierarchical fusion strategy that adaptively fuses representations from different layers. Three fusion approaches are used: (*i*) Concat Fusion, (*ii*) Max Fusion, and (*iii*) Gate Fusion. The proposed method reduces the similarity between tokens and outperforms BERT baseline on the GLUE (Wang et al., 2018a), SWAG (Zellers et al., 2018) and SQuAD (Rajpurkar et al., 2016; 2018) data sets.

In summary, the contributions of this paper are as follows: (*i*) We develop the relationship between self-attention and graph for a better understanding of over-smoothing in BERT. (*ii*) We provide theoretical analysis on over-smoothing in the BERT model, and empirically verify the theoretical results. (*iii*) We propose hierarchical fusion strategies that adaptively combine different layers to alleviate over-smoothing. Extensive experimental results verify our methods' effectiveness.

## 2 RELATED WORK

### 2.1 TRANSFORMER BLOCK AND SELF-ATTENTION

Transformer block is a basic component in Transformer model (Vaswani et al., 2017). Each Transformer block consists of a self-attention layer and a feed-forward layer. Let $\boldsymbol{X} \in \mathbb{R}^{n \times d}$ be the input to a Transformer block, where $n$ is the number of input tokens and $d$ is the embedding size. The self-attention layer output can be written as:

$$Attn(\boldsymbol{X}) = \boldsymbol{X} + \sum_{k=1}^{h} \sigma(\boldsymbol{X}\boldsymbol{W}_k^Q(\boldsymbol{X}\boldsymbol{W}_k^K)^\top)\boldsymbol{X}\boldsymbol{W}_k^V\boldsymbol{W}_k^{O\top} = \boldsymbol{X} + \sum_{k=1}^{h} \hat{\boldsymbol{A}}_k \boldsymbol{X}\boldsymbol{W}_k^{VO}, \quad (1)$$

where $h$ is the number of heads, $\sigma$ is the softmax function, and $\boldsymbol{W}_k^Q, \boldsymbol{W}_k^K, \boldsymbol{W}_k^V, \boldsymbol{W}_k^O \in \mathbb{R}^{d \times d_h}$ (where $d_h = d/h$ is the dimension of a single-head output) are weight matrices for the query, key, value, and output, respectively of the $k$th head. In particular, the self-attention matrix

$$\hat{\boldsymbol{A}} = \sigma(\boldsymbol{X}\boldsymbol{W}^Q(\boldsymbol{X}\boldsymbol{W}^K)^\top) = \sigma(\boldsymbol{Q}\boldsymbol{K}^\top) \quad (2)$$

in (1) plays a key role in the self-attention layer (Park et al., 2019; Gong et al., 2019; Kovaleva et al., 2019). As in (Yun et al., 2020; Shi et al., 2021; Dong et al., 2021), we drop the scale product $1/\sqrt{d_h}$ to simplify analysis.

The feed-forward layer usually has two fully-connected (FC) layers with residual connection:

$$FF(\boldsymbol{X}) = Attn(\boldsymbol{X}) + ReLU(Attn(\boldsymbol{X})\boldsymbol{W}_1 + \boldsymbol{b}_1)\boldsymbol{W}_2 + \boldsymbol{b}_2,$$

where $\boldsymbol{W}_1 \in \mathbb{R}^{d \times d_{\text{ff}}}, \boldsymbol{W}_2 \in \mathbb{R}^{d_{\text{ff}} \times d}$ ($d_{\text{ff}}$ is the size of the intermediate layer) are the weight matrices, and $\boldsymbol{b}_1, \boldsymbol{b}_2$ are the biases. Two layer normalization (Ba et al., 2016) operations are performed after the self-attention layer and fully-connected layer, respectively.

### 2.2 OVER-SMOOTHING

In graph neural networks, over-smoothing refers to the problem that the performance deteriorates as representations of all the nodes become similar (Li et al., 2018; Xu et al., 2018; Huang et al., 2020). Its main cause is the stacked aggregation layer using the same adjacency matrix. Recently, several approaches have been proposed to alleviate the over-smoothing problem. Xu et al. (2018) propose a jumping knowledge network for better structure-aware representation, which flexibly

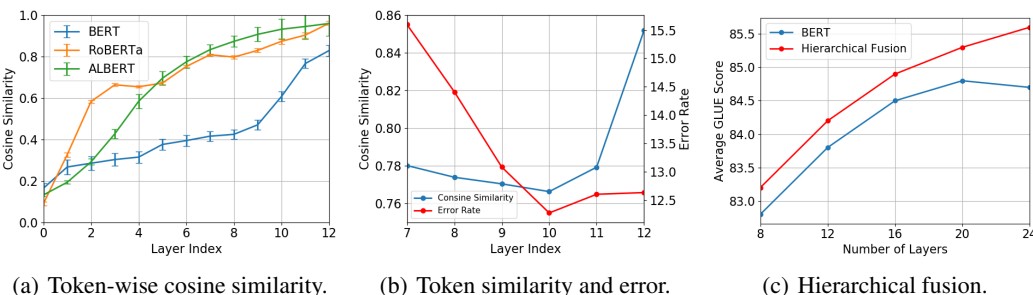

(a) Token-wise cosine similarity.  (b) Token similarity and error.  (c) Hierarchical fusion.

Figure 1: Over-smoothing in BERT models.

leverages different neighborhood ranges. ResGCN (Li et al., 2019) adapts the residual connection and dilated convolution in the graph convolutional network (GCN), and successfully scales the GCN to 56 layers. Zhao & Akoglu (2020) propose PairNorm, a novel normalization layer, that prevents node embeddings from becoming too similar. DropEdge (Rong et al., 2020; Huang et al., 2020) randomly removes edges from the input graph at each training epoch, and reduces the effect of over-smoothing.

Unlike graph neural networks, over-smoothing in Transformer-based architectures has not been discussed in detail. Dong et al. (2021) introduce the "token-uniformity" problem for self-attention, and show that skip connections and multi-layer perceptron can mitigate this problem. However, Gong et al. (2021) still observe over-smoothing on the Vision Transformers (Dosovitskiy et al., 2021).

## 3 DOES OVER-SMOOTHING EXIST IN BERT?

In this section, we first explore the existence of over-smoothing in BERT, by measuring the similarity between tokens in each Transformer layer. Specifically, we use the token-wise cosine similarity (Gong et al., 2021) as our similarity measure:

$$\text{CosSim} = \frac{1}{n(n-1)} \sum_{i \neq j} \frac{\boldsymbol{h}_i^\top \boldsymbol{h}_j}{\|\boldsymbol{h}_i\|_2 \|\boldsymbol{h}_j\|_2},$$

where $n$ is the number of tokens, $\boldsymbol{h}_i$ and $\boldsymbol{h}_j$ are two representations of different tokens, and $\| \cdot \|_2$ is the Euclidean norm. Following Dong et al. (2021), we use WikiBio (Lebret et al., 2016) as input to the following Transformer-based models fine-tuned on the SQuAD data set (Rajpurkar et al., 2018): (*i*) BERT (Devlin et al., 2019), (*ii*) RoBERTa (Liu et al., 2019) and (*iii*) ALBERT (Lan et al., 2020).[1] For comparison, all three models are stacked with 12 blocks. We calculate each *CosSim* for each data sample and show the average and standard derivation of *CosSim* values over all WikiBio data.

In the figures, layer 0 represents original input token representation, and layer 1-12 represents the corresponding transformer layers. As shown in Figure 1(a), the original token representations are different from each other, while token similarities are high in the last layer. For instance, the average token-wise cosine similarity of the last layer of ALBERT and RoBERTa are both larger than 90%.

To illustrate the relationship between "over-thinking" and "over-smoothing", we compare the token-wise cosine similarity at each layer with the corresponding error rate. As for the corresponding error rate of layer $i$, we use the representations from layer $i$ as the final output and fine-tune the classifier. Following Zhou et al. (2020), we experiment with ALBERT (Lan et al., 2020) fine-tuned on the MRPC data set (Dolan & Brockett, 2005) and use their error rate results for convenience. As shown in Figure 1(b), layer 10 has the lowest cosine similarity and error rate. At layers 11 and 12, the tokens have larger cosine similarities, making them harder to distinguish and resulting in the performance drop. Thus, "over-thinking" can be explained by "over-smoothing".

A direct consequence of over-smoothing is that the performance cannot be improved when the model gets deeper, since the individual tokens are no longer distinguishable. To illustrate this, we increase the number of layers in BERT to 24 while keeping the other settings. As shown in Figure 1(c), the

---

[1]Our implementation is based on the HuggingFace's Transformers library (Wolf et al., 2020).

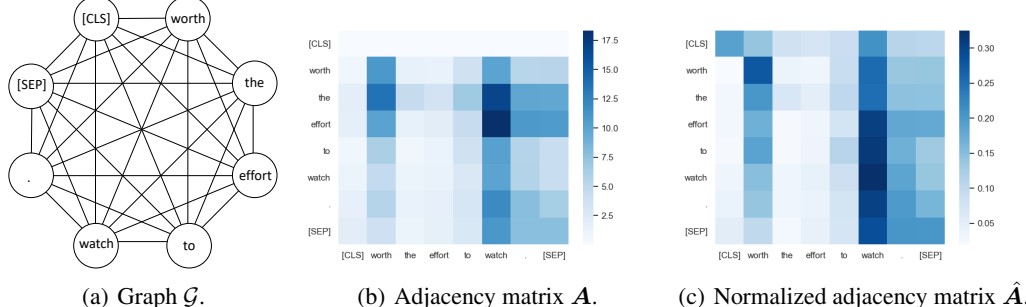

(a) Graph $\mathcal{G}$.    (b) Adjacency matrix $\boldsymbol{A}$.    (c) Normalized adjacency matrix $\hat{\boldsymbol{A}}$.

Figure 2: Illustration of self-attention and the corresponding graph $\mathcal{G}$. For simplicity, we drop the self-loops in $\mathcal{G}$.

performance of vanilla BERT cannot improve as the model gets deeper. In contrast, the proposed hierarchical fusion (as will be discussed in Section 6) consistently outperforms the baseline, and has better and better performance as the model gets deeper. Based on these observations, we conclude that the over-smoothing problem still exists in BERT.

## 4 RELATIONSHIP BETWEEN SELF-ATTENTION AND GRAPH

Since over-smoothing is first discussed in the graph neural network literature (Li et al., 2018; Zhao & Akoglu, 2020), we attempt to understand its cause from a graph perspective in this section.

### 4.1 SELF-ATTENTION VS RESGCN

Given a Transformer block, construct a weighted graph $\mathcal{G}$ with the input tokens as nodes and $\exp(\boldsymbol{Q}_i^\top \boldsymbol{K}_j)$ as the $(i,j)$th entry of its adjacency matrix $\boldsymbol{A}$. By rewriting the self-attention matrix $\hat{\boldsymbol{A}}$ in (2) as $\hat{A}_{i,j} = \sigma(\boldsymbol{Q}\boldsymbol{K}^\top)_{i,j} = \exp(\boldsymbol{Q}_i^\top \boldsymbol{K}_j)/\sum_l \exp(\boldsymbol{Q}_i^\top \boldsymbol{K}_l)$, $\hat{\boldsymbol{A}}$ can thus be viewed as $\mathcal{G}$'s normalized adjacency matrix (Von Luxburg, 2007). In other words, $\hat{\boldsymbol{A}} = \boldsymbol{D}^{-1}\boldsymbol{A}$, where $\boldsymbol{D} = \mathrm{diag}(d_1, d_2, \ldots, d_n)$ and $d_i = \sum_j A_{i,j}$. Figure 2 shows an example for the sentence "worth the effort to watch." from the SST-2 data set (Socher et al., 2013) processed by BERT.

Note that graph convolutional network combined with residual connections (ResGCN) (Kipf & Welling, 2017) can be expressed as follows.

$$ResGCN(\boldsymbol{X}) = \boldsymbol{X} + ReLU(\boldsymbol{D}^{-1/2}\boldsymbol{A}\boldsymbol{D}^{-1/2}\boldsymbol{X}\boldsymbol{W}) = \boldsymbol{X} + ReLU(\hat{\boldsymbol{A}}\boldsymbol{X}\boldsymbol{W}), \tag{3}$$

which has the similar form with the self-attention layer in Eq. (1). By comparing self-attention module with ResGCN, we have the following observations: (i) Since $A_{i,j} \neq A_{j,i}$ in general, $\mathcal{G}$ in self-attention is a directed graph; (ii) $\hat{\boldsymbol{A}} = \boldsymbol{D}^{-1}\boldsymbol{A}$ in self-attention is the random walk normalization (Chung & Graham, 1997), while GCN usually uses the symmetric normalization version $\hat{\boldsymbol{A}} = \boldsymbol{D}^{-1/2}\boldsymbol{A}\boldsymbol{D}^{-1/2}$; (iii) The attention matrices constructed at different Transformer layers are different, while in typical graphs, the adjacency matrices are usually static.

### 4.2 UNSHARED ATTENTION MATRIX VS SHARED ATTENTION MATRIX

As discussed in Section 2.2, over-smoothing in graph neural networks is mainly due to the repeated aggregation operations using the same adjacency matrix. To compare the self-attention matrices ($\hat{\boldsymbol{A}}$'s) at different Transformer layers, we first flatten the multi-head attention and then measure the cosine similarity between $\hat{\boldsymbol{A}}$'s at successive layers. Experiment is performed with BERT (Devlin et al., 2019), RoBERTa (Liu et al., 2019) and ALBERT (Lan et al., 2020) on the WikiBio data set (Lebret et al., 2016).

Table 1: Performance (%) on the GLUE development set by the original BERT (top row) and various BERT variants with different degrees of self-attention matrix sharing. Numbers in parentheses are the layers that share the self-attention matrix (e.g., BERT (1-12) means that the $\hat{A}$'s from layers 1-12 are shared). The last column shows the FLOPs in the self-attention modules.

| | MNLI (m/mm) | QQP | QNLI | SST-2 | COLA | STS-B | MRPC | RTE | Average | FLOPs |
|---|---|---|---|---|---|---|---|---|---|---|
| BERT | 85.4/85.8 | 88.2 | 91.5 | 92.9 | 62.1 | 88.8 | 90.4 | 69.0 | 83.8 | 2.7G |
| BERT (11-12) | 84.9/85.0 | 88.1 | 91.0 | 93.0 | 62.3 | 89.7 | 91.1 | 70.8 | 84.0 | 2.4G |
| BERT (9-12) | 85.3/85.1 | 88.1 | 90.1 | 92.9 | 62.6 | 89.3 | 91.2 | 68.5 | 83.7 | 2.1G |
| BERT (7-12) | 84.2/84.8 | 88.0 | 90.6 | 92.1 | 62.7 | 89.2 | 90.5 | 68.2 | 83.4 | 1.8G |
| BERT (5-12) | 84.0/84.3 | 88.0 | 89.7 | 92.8 | 64.1 | 89.0 | 90.3 | 68.2 | 83.4 | 1.5G |
| BERT (3-12) | 82.5/82.4 | 87.5 | 88.6 | 91.6 | 57.0 | 87.9 | 88.4 | 65.7 | 81.3 | 1.2G |
| BERT (1-12) | 81.3/81.7 | 87.3 | 88.5 | 92.0 | 57.7 | 87.4 | 87.5 | 65.0 | 80.9 | 1.1G |

Figure 3 shows the cosine similarities obtained. As can be seen, the similarities at the last few layers are high,[2] while those at the first few layers are different from each other. In other words, the attention patterns at the first few layers are changing, and become stable at the upper layers.

In the following, we focus on BERT and explore how many layers can share the same self-attention matrix. Note that this is different from ALBERT, which shares model parameters instead of attention matrices. Results are shown in Table 1. As can be seen, sharing attention matrices among the last $8$ layers (i.e., layers 5-12) does not harm model performance. This is consistent with the observation in Figure 3. Note that sharing attention matrices not only reduces the number of parameters in the self-attention module, but also makes the model more efficient by reducing the computations during training and inference. As shown in Table 1, BERT (5-12) reduces $44.4\%$ FLOPs in the self-attention modules compared with the vanilla BERT, while still achieving comparable average GLUE scores.

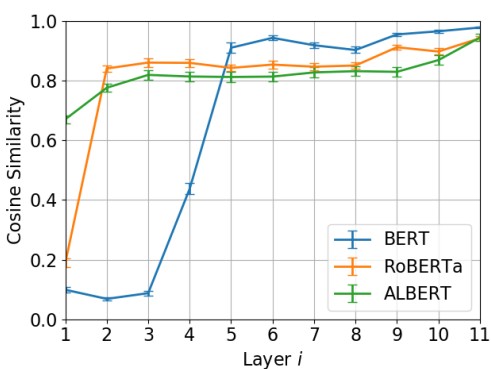

Figure 3: Consine similarity between the attention matrices $\hat{A}$'s at layer $i$ and its next higher layer.

## 5    OVER-SMOOTHING IN BERT

In this section, we analyze the over-smoothing problem in BERT theoretically, and then verify the result empirically.

### 5.1    THEORETICAL ANALYSIS

Our analysis is based on matrix projection. We define a subspace $\mathcal{M}$, in which each row vector of the element in this subspace is identical.

**Definition 1.** *Define $\mathcal{M} := \{Y \in \mathbb{R}^{n \times d} | Y = eC, C \in \mathbb{R}^{1 \times d}\}$ as a subspace in $\mathbb{R}^{n \times d}$, where $e = [1, 1, \ldots, 1]^\top \in \mathbb{R}^{n \times 1}$, $n$ is the number of tokens and $d$ is the dimension of token representation.*

Each $Y$ in subspace $\mathcal{M}$ suffers from the over-smoothing issue since the representation of each token is $C$, which is the same with each other. We define the distance between matrix $H \in \mathbb{R}^{n \times d}$ and $\mathcal{M}$ as $d_{\mathcal{M}}(H) := \min_{Y \in \mathcal{M}} \|H - Y\|_F$, where $\| \cdot \|_F$ is the Frobenius norm. Next, we investigate the distance between the output of layer $l$ and subspace $\mathcal{M}$. We have the following Lemma.

---

[2]For example, in BERT, the attention matrices $\hat{A}$'s for the last 8 layers are very similar.

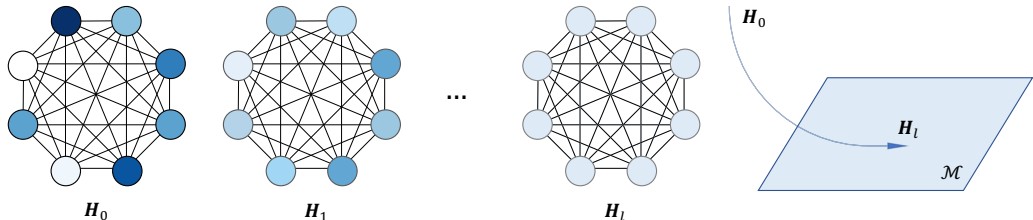

Figure 4: The illustration of over-smoothing problem. Recursively, $\boldsymbol{H}_l$ will converge to subspace $\mathcal{M}$ where representation of each token is identical.

**Lemma 1.** *For self-attention matrix $\hat{\boldsymbol{A}}$, any $\boldsymbol{H}, \boldsymbol{B} \in \mathbb{R}^{n \times d}$ and $\alpha_1, \alpha_2 \geq 0$, we have:*

$$d_{\mathcal{M}}(\boldsymbol{H}\boldsymbol{W}) \leq s d_{\mathcal{M}}(\boldsymbol{H}), \tag{4}$$

$$d_{\mathcal{M}}(ReLU(\boldsymbol{H})) \leq d_{\mathcal{M}}(\boldsymbol{H}), \tag{5}$$

$$d_{\mathcal{M}}(\alpha_1 \boldsymbol{H} + \alpha_2 \boldsymbol{B}) \leq \alpha_1 d_{\mathcal{M}}(\boldsymbol{H}) + \alpha_2 d_{\mathcal{M}}(\boldsymbol{B}), \tag{6}$$

$$d_{\mathcal{M}}(\hat{\boldsymbol{A}}\boldsymbol{H}) \leq \sqrt{\lambda_{\max}} d_{\mathcal{M}}(\boldsymbol{H}), \tag{7}$$

*where $\lambda_{\max}$ is the largest eigenvalue of $\hat{\boldsymbol{A}}^\top (\boldsymbol{I} - \boldsymbol{e}\boldsymbol{e}^\top)\hat{\boldsymbol{A}}$ and $s$ is the largest singular value of $\boldsymbol{W}$.*

Using Lemma 1, we have the following Theorem.

**Theorem 2.** *For a BERT block with $h$ heads, we have*

$$d_{\mathcal{M}}(\boldsymbol{H}_{l+1}) \leq v d_{\mathcal{M}}(\boldsymbol{H}_l), \tag{8}$$

*where $v = (1 + s^2)(1 + \sqrt{\lambda}hs)/(\sigma_1 \sigma_2)$, $s > 0$ is the largest element of all singular values of all $\boldsymbol{W}_l$, $\lambda$ is the largest eigenvalue of all $\hat{\boldsymbol{A}}^\top (\boldsymbol{I} - \boldsymbol{e}\boldsymbol{e}^\top)\hat{\boldsymbol{A}}$ for each self-attention matrix $\hat{\boldsymbol{A}}$, and $\sigma_1$, $\sigma_2$ are the minimum standard deviation for two layer normalization operations.*

Proof is in Appendix A. Theorem 2 shows that if $v < 1$ (i.e., $\sigma_1 \sigma_2 > (1 + s^2)(1 + \sqrt{\lambda}hs)$), the output of layer $l + 1$ will be closer to $\mathcal{M}$ than the output of layer $l$. An illustration of Theorem 2 is shown in Figure 4. $\boldsymbol{H}_0$ is the graph corresponding to the input layer. Initially, the token representations are very different (indicated by the different colors of the nodes). Recursively, $\boldsymbol{H}_l$ will converge towards to subspace $\mathcal{M}$ if $v < 1$ and all representations are the same, resulting in over-smoothing.

**Remark** Though we only focus on the case $v < 1$, over-smoothing may still exist if $v \geq 1$.

As can be seen, layer normalization plays an important role for the convergence rate $v$. Interestingly, Dong et al. (2021) claim that layer normalization plays no roles for token uniformity, which seems to conflict with the conclusion in Theorem 2. However, note that the matrix rank cannot indicate similarity between tokens completely because matrix rank is discrete while similarity is continuous. For instance, given two token embeddings $\boldsymbol{h}_i$ and $\boldsymbol{h}_j$, the matrix $[\boldsymbol{h}_i, \boldsymbol{h}_j]^\top$ has rank 2 only if $\boldsymbol{h}_i \neq \boldsymbol{h}_j$. In contrast, the consine similarity between tokens is $\frac{\boldsymbol{h}_i^\top \boldsymbol{h}_j}{\|\boldsymbol{h}_i\|_2 \|\boldsymbol{h}_j\|_2}$.

As discussed in Section 4.1, GCN use the symmetric normalization version $\hat{\boldsymbol{A}} = \boldsymbol{D}^{-1/2}\boldsymbol{A}\boldsymbol{D}^{-1/2}$, resulting in the target subspace $\mathcal{M}' := \{\boldsymbol{Y} \in \mathbb{R}^{n \times d} | \boldsymbol{Y} = \boldsymbol{D}^{1/2}\boldsymbol{e}\boldsymbol{C}, \boldsymbol{C} \in \mathbb{R}^{1 \times d}\}$ is dependent with adjacent matrix (Huang et al., 2020). In contrast, our subspace $\mathcal{M}$ is independent of $\hat{\boldsymbol{A}}$ thanks to its random walk normalization. Thus, Theorem 2 can be applied to the vanilla BERT even though its attention matrix $\hat{\boldsymbol{A}}$ is not similar.

## 5.2 EMPIRICAL VERIFICATION

Theorem 2 illustrates that the magnitude of $\sigma_1 \sigma_2$ is important for over-smoothing issue. If $\sigma_1 \sigma_2 > (1 + s^2)(1 + \sqrt{\lambda}hs)$, the output will be closer to subspace $\mathcal{M}$ suffered from over-smoothing. Since $s$ is usually small due to the $\ell_2$-penalty during training (Huang et al., 2020), we neglect the effect of $s$ and compare $\sigma_1 \sigma_2$ with 1 for simplicity. To verify the theoretical results, we visualize $\sigma_1 \sigma_2$ in different fine-tuned BERT models. Specifically, we take the development set data of STS-B (Cer et al., 2017),

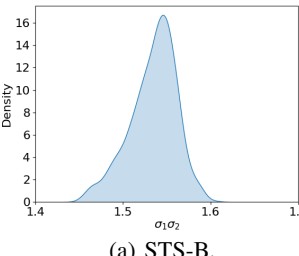 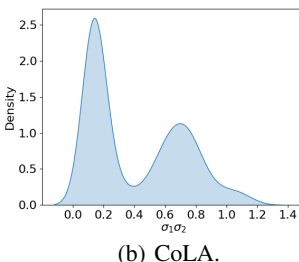 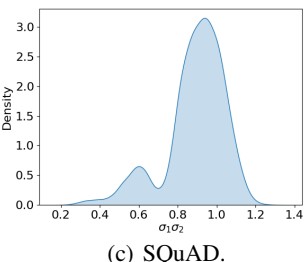

(a) STS-B.          (b) CoLA.          (c) SQuAD.

Figure 5: The estimated distribution of $\sigma_1\sigma_2$ for different fine-tuned models.

CoLA (Warstadt et al., 2019), SQuAD (Rajpurkar et al., 2016) as input to the fine-tuned models and visualize the distribution of $\sigma_1\sigma_2$ at the last layer using kernel density estimation (Rosenblatt, 1956).

Results are shown in Figure 5. As can be seen, the distributions of $\sigma_1\sigma_2$ can be very different across data sets. For STS-B (Cer et al., 2017), $\sigma_1\sigma_2$ of all data is larger than 1, which means that over-smoothing is serious for this data set. For CoLA (Warstadt et al., 2019) and SQuAD (Rajpurkar et al., 2016), there also exists a fraction of samples satisfying $\sigma_1\sigma_2 > 1$.

## 6 METHOD

From our proof in Appendix A, we figure out that the main reason is the post-normalization scheme in BERT. In comparison, to train a 1000-layer GCN, Li et al. (2021) instead apply pre-normalization with skip connections to ensure $v > 1$. However, the performance of pre-normalization is not better than post-normalization for layer normalization empirically (He et al., 2021). In this section, we preserve the post-normalization scheme and propose a hierarchical fusion strategy to alleviate the over-smoothing issue. Specifically, since only deep layers suffer from the over-smoothing issue, we allow the model select representations from both shallow layers and deep layers as final output.

### 6.1 HIERARCHICAL FUSION STRATEGY

**Concat Fusion** We first consider a simple and direct layer-wise Concat Fusion approach. Considering a $L$-layer model, we first concatenate the representations $\boldsymbol{H}_k$ from each layer $k$ to generate a matrix $[\boldsymbol{H}_1, \boldsymbol{H}_2, \ldots, \boldsymbol{H}_L]$ and then apply a linear mapping to generate the final representation $\sum_{k=1}^{L} \alpha_k \boldsymbol{H}_k$. Here $\{\alpha_k\}$ are model parameters independent with inputs. Since this scheme requires preserving feature maps from all layers, the memory cost will be huge as the model gets deep.

**Max Fusion** Inspired by the idea of the widely adopted max-pooling mechanism, we construct the final output by taking the maximum value across all layers for each dimension of the representation. Max Fusion is an adaptive fusion mechanism since the model can dynamically decide the important layer for each element in the representation. Max Fusion is the most flexible strategy, since it does not require learning any additional parameters and is more efficient in terms of speed and memory.

**Gate Fusion** Gate mechanism is commonly used for information propagation in natural language processing field (Cho et al., 2014). To exploit the advantages from different semantic levels, we propose a vertical gate fusion module, which predicts the respective importance of token-wise representations from different layers and aggregate them adaptively. Given token representations $\{\boldsymbol{H}_k^t\}$, where $t$ denotes the token index and $k$ denotes the layer index, the final representation for token $t$ is calculated by $\sum_{k=1}^{L} I_k^t \cdot \boldsymbol{H}_k^t$, where $I_1^t, I_2^t, \ldots, I_L^t = \text{softmax}(g(\boldsymbol{H}_1^t), g(\boldsymbol{H}_2^t), \ldots, g(\boldsymbol{H}_L^t))$. Here $L$ is the number of layers and the gate function $g(\cdot)$ is a fully-connected (FC) layer, which relies on the word representation itself in respective layers to predict its importance scores. The weights of the gate function $g(\cdot)$ are shared across different layers.

Even though Concat Fusion and Max Fusion have been investigated in the graph field (Xu et al., 2018), their effectiveness for pre-trained language model have not yet been explored. Besides, since the *layer-wise* Concat Fusion and *element-wise* Max Fusion lack the ability to generate token representations according to each token's specificity, we further propose the *token-wise* Gate Fusion for adapting fusion to the language scenario.

Table 2: Performance (in %) of the various BERT variants on the GLUE development data set.

| | MNLI (m/mm) | QQP | QNLI | SST-2 | COLA | STS-B | MRPC | RTE | Average |
|---|---|---|---|---|---|---|---|---|---|
| BERT | 85.4/85.8 | 88.2 | 91.5 | 92.9 | 62.1 | 88.8 | 90.4 | 69.0 | 83.8 |
| BERT (concat) | 85.3/85.4 | 87.8 | 91.8 | 93.8 | 65.1 | 89.8 | 91.3 | 71.1 | 84.6 |
| BERT (max) | 85.3/85.6 | 88.5 | 92.0 | 93.7 | 64.6 | 90.3 | 91.7 | 71.5 | 84.7 |
| BERT (gate) | 85.4/85.7 | 88.4 | 92.3 | 93.9 | 64.0 | 90.3 | 92.0 | 73.9 | **85.1** |
| ALBERT | 81.6/82.2 | 85.6 | 90.7 | 90.3 | 50.8 | 89.4 | 91.3 | 75.5 | 81.8 |
| ALBERT (concat) | 82.8/82.8 | 86.7 | 90.9 | 90.7 | 48.7 | 89.7 | 91.5 | 76.5 | 82.3 |
| ALBERT (max) | 82.5/82.8 | 86.9 | 91.1 | 90.7 | 50.5 | 89.6 | 92.6 | 77.3 | 82.6 |
| ALBERT (gate) | 83.0/83.7 | 87.0 | 90.9 | 90.4 | 51.3 | 90.0 | 92.4 | 76.2 | **82.7** |

## 6.2 EXPERIMENT RESULTS

The BERT model is stacked with 12 Transformer blocks (Section 2.1) with the following hyper-parameters: number of tokens $n = 128$, number of self-attention heads $h = 12$, and hidden layer size $d = 768$. As for the feed-forward layer, we set the filter size $d_{\text{ff}}$ to 3072 as in Devlin et al. (2019). All experiments are performed on NVIDIA Tesla V100 GPUs.

### 6.2.1 DATA AND SETTINGS

**Pre-training** For the setting in pre-training phase, we mainly follows BERT paper (Devlin et al., 2019). Our pre-training tasks are vanilla masked language modeling (MLM) and next sentence prediction (NSP). The pre-training datasets are English BooksCorpus (Zhu et al., 2015) and Wikipedia (Devlin et al., 2019) (16G in total). The WordPiece embedding (Wu et al., 2016) and the dictionary containing $30,000$ tokens in (Devlin et al., 2019) are still used in our paper. To pre-process text, we use the special token [CLS] as the first token of each sequence and [SEP] to separate sentences in a sequence. The pre-training is performed for 40 epochs.

**Fine-tuning** In the fine-tuning phase, we perform downstream experiments on the GLUE (Wang et al., 2018a), SWAG (Zellers et al., 2018) and SQuAD (Rajpurkar et al., 2016; 2018) benchmarks. GLUE is a natural language understanding benchmark, which includes three categories tasks: (i) single-sentence tasks (CoLA and SST-2); (ii) similarity and paraphrase tasks (MRPC, QQP and STS-B); (iii) inference tasks (MNLI, QNLI and RTE). For MNLI task, we experiment on both the matched (MNLI-m) and mismatched (MNLI-mm) versions. The SWAG data set is for grounded commonsense inference, while SQuAD is a task for question answering. In SQuAD v1.1 (Rajpurkar et al., 2016), the answers are included in the context. SQuAD v2.0 (Rajpurkar et al., 2018) is more challenge than SQuAD v1.0, in which some answers are not included in the context. Following BERT (Devlin et al., 2019), we report accuracy for MNLI, QNLI, RTE, SST-2 tasks, F1 score for QQP and MRPC, Spearman correlation for STS-B, and Matthews correlation for CoLA. For SWAG task, we use accuracy for evaluation. For SQuAD v1.1 and v2.0, we report the Exact Match (EM) and F1 scores. Descriptions of the data sets and details of other hyper-parameter settings are in Appendix B and in Appendix C, respectively.

### 6.2.2 RESULTS AND ANALYSIS

Since BERT (Devlin et al., 2019) and RoBERTa (Liu et al., 2019) share the same architecture and the only difference is data resource and training steps, here we mainly evaluate our proposed method on BERT and ALBERT (Lan et al., 2020). Results on the GLUE benchmark are shown in Table 2, while results on SWAG and SQuAD are illustrated in Table 3. For SQuAD task, in contrast to BERT which (Devlin et al.,

Table 3: Performance (in %) on the SWAG and SQuAD development sets.

| | SWAG | SQuAD v1.1 | | SQuAD v2.0 | |
|---|---|---|---|---|---|
| | acc | EM | F1 | EM | F1 |
| BERT | 81.6 | 79.7 | 87.1 | 72.9 | 75.5 |
| BERT (concat) | 82.0 | 80.2 | 87.8 | **74.1** | 77.0 |
| BERT (max) | 81.9 | 80.1 | 87.6 | 73.6 | 76.6 |
| BERT (gate) | **82.1** | **80.7** | **88.0** | 73.9 | **77.3** |

2019) utilize the augmented training data during fine-tuning phase, we only fine-tune our model on the standard SQuAD data set. As can be seen, our proposed fusion strategies also perform better than baselines on various tasks consistently.

Following the previous over-smoothing measure, we visualize the token-wise cosine similarity in each layer. Here we perform visualization on the same data sets as Section 5.2 and the results are shown in Figure 6. For all three data sets, the cosine similarity has a drop in the last layer compared

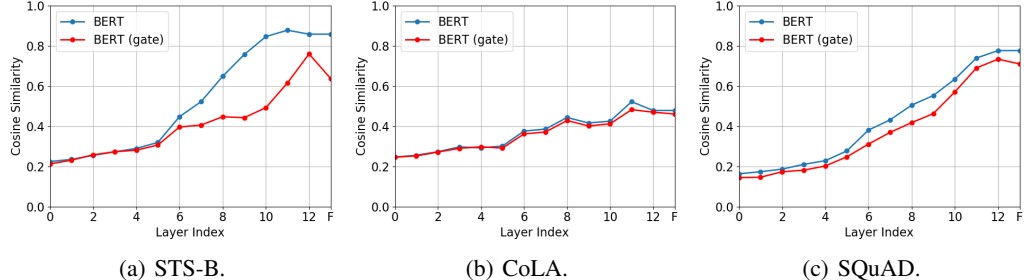

(a) STS-B.    (b) CoLA.    (c) SQuAD.

Figure 6: The token-wise similarity comparison between BERT and BERT with gate fusion. Here F means the final output, which is the fusion results for our approach.

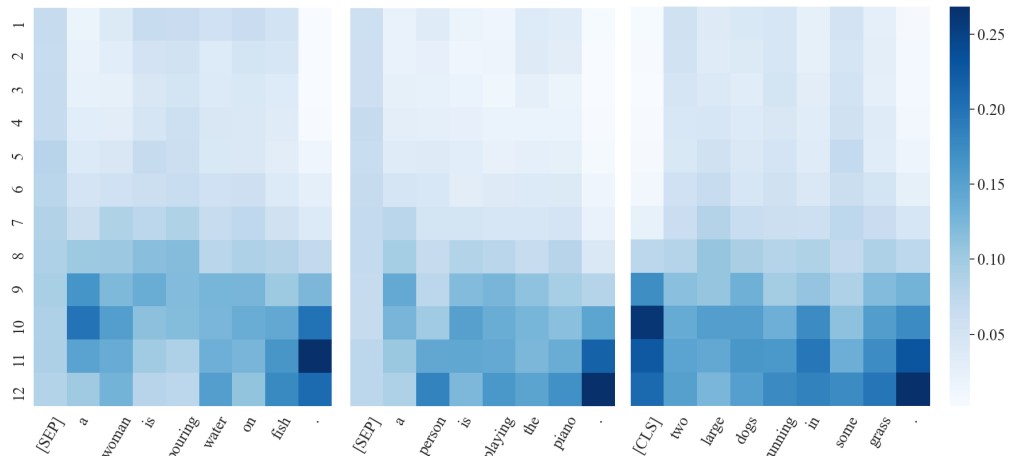

Figure 7: Visualization of importance weights of gate fusion on different layers.

with baseline. It's remarkable that the similarity drop is the most obvious in STS-B (Cer et al., 2017), which is consistent with our empirical verification that STS-B's $\sigma_1\sigma_2$ is the largest in Section 5.2. Since the representation of tokens from prior layers is not similar with each other, our fusion method alleviates the over-smoothing issue and improve the model performance at the same time.

To study the dynamic weights of fusion gate strategy, we visualize the importance weight $I_k^t$ for each token $t$ and for each layer $k$. We randomly select three samples and the visualization results are illustrated in Figure 7. Note that our gate strategy will reduce to vanilla model if representation from the last layer is selected for each token. As can be seen, the weight distribution of different tokens is adaptively decided, illustrating that the vanilla BERT stacks model is not the best choice for all tokens. The keywords which highly affect meaning of sentences (i.e. *"women", "water", "fish"*) are willing to obtain more semantic representations from the deep layer, while for some simple words which appear frequently (i.e. *"a", "is"*), the features in shallow layers are preferred.

# 7    CONCLUSION

In this paper, we revisit the over-smoothing problem in BERT models. Since this issue has been detailed discuss in graph learning field, we firstly establish the relationship between BERT and graph for inspiration, and find out that self-attention matrix can be shared among last few blocks without performance drop. Inspired by over-smoothing discussion in graph convolutional network, we provide some theoretical analysis for BERT models and figure out the importance of layer normalization. Specifically, if the standard derivation of layer normalization is sufficiently large, the output will converge towards to a low-rank subspace. To alleviate the over-smoothing problem, we also propose a hierarchical fusion strategy to combine representations from different layers adaptively. Extensive experiment results on various data sets illustrate the effect of our fusion methods.

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

# A  PROOF

**Lemma 1.** *For self-attention matrix $\hat{A}$, any $H, B \in \mathbb{R}^{n \times d}$ and $\alpha_1, \alpha_2 \geq 0$, we have:*

$$d_{\mathcal{M}}(HW) \leq sd_{\mathcal{M}}(H), \tag{4}$$

$$d_{\mathcal{M}}(ReLU(H)) \leq d_{\mathcal{M}}(H), \tag{5}$$

$$d_{\mathcal{M}}(\alpha_1 H + \alpha_2 B) \leq \alpha_1 d_{\mathcal{M}}(H) + \alpha_2 d_{\mathcal{M}}(B), \tag{6}$$

$$d_{\mathcal{M}}(\hat{A}H) \leq \sqrt{\lambda_{\max}} d_{\mathcal{M}}(H), \tag{7}$$

*where $\lambda_{\max}$ is the largest eigenvalue of $\hat{A}^\top(I - ee^\top)\hat{A}$ and $s$ is the largest singular value of $W$.*

*Proof.* Here we only prove the last inequality (7), as the inequity is different from the theories in GCN since $\hat{A}$ is not symmetric and shared in Transformer architecture. For the first three inequalities, we refer to Oono & Suzuki (2020) and Huang et al. (2020).

Write $HH^\top = Q\Omega Q^\top$ for the eigin-decomposition of $HH^\top$, where $Q = [q_1, q_2, \ldots, q_n]$ is the orthogonal and $\Omega = \text{diag}(\omega_1, \ldots, \omega_n)$ with all $\omega_i \geq 0$. Recall $e = n^{-1/2}[1, 1, \ldots, 1]^\top \in \mathbb{R}^{n \times 1}$.

Note that

$$
\begin{aligned}
d_{\mathcal{M}}(\hat{A}H)^2 &= \|(I - ee^\top)\hat{A}H\|_F^2 \\
&= tr\{(I - ee^\top)\hat{A}HH^\top\hat{A}^\top(I - ee^\top)\} \\
&= \sum_{i=1}^n \omega_i q_i^\top \hat{A}^\top(I - ee^\top)\hat{A}q_i.
\end{aligned}
$$

Since matrix $\hat{A}^\top(I - ee^\top)\hat{A}$ is positive semidefinite, its all eigenvalues are non-negative. Let $\lambda_{\max}$ be the largest eigenvalue of $\hat{A}^\top(I - ee^\top)\hat{A}$. Consider

$$\lambda_{\max} d_{\mathcal{M}}(H)^2 - d_{\mathcal{M}}(\hat{A}H)^2 = \sum_{i=1}^n \omega_i q_i^\top \{\lambda_{\max}(I - ee^\top) - \hat{A}^\top(I - ee^\top)\hat{A}\}q_i.$$

Let $\Sigma = \lambda_{\max}(I - ee^\top) - \hat{A}^\top(I - ee^\top)\hat{A}$.

Note that $\hat{A} = D^{-1}A$ is a stochastic matrix, we have $\hat{A}e = e$. Thus, $\hat{A}^\top(I - ee^\top)\hat{A}$ has an eigenvalue 0 and corresponding eigenvecter $e$. Let $f_i$ be a normalised eigenvector of $\hat{A}^\top(I - ee^\top)\hat{A}$ orthogonal to $e$, and $\lambda$ be its corresponding eigenvalue. Then we have

$$e^\top \Sigma e = 0,$$

$$f_i^\top \Sigma f_i = \lambda_{\max} - \lambda \geq 0.$$

It follows that $d_{\mathcal{M}}(\hat{A}H)^2 \leq \lambda_{\max} d_{\mathcal{M}}(H)^2$.   $\square$

**Discussion** Assume further that $\hat{A}$ is doubly stochastic (so that $\hat{A}^\top e = e$) with positive entries. Then by Perron–Frobenius theorem (Gantmakher, 2000), $\hat{A}^\top\hat{A}$ has a maximum eigenvalue 1 with associated eigenvector $e$ as well. In this case, the matrix $\hat{A}^\top(I - ee^\top)\hat{A} = \hat{A}^\top\hat{A} - ee^\top$ has a maximum eigenvalue $\lambda_{max} < 1$.

**Theorem 2.** *For a BERT block with $h$ heads, we have*

$$d_{\mathcal{M}}(H_{l+1}) \leq vd_{\mathcal{M}}(H_l), \tag{8}$$

*where $v = (1 + s^2)(1 + \sqrt{\lambda}hs)/(\sigma_1\sigma_2)$, $s > 0$ is the largest element of all singular values of all $W_l$, $\lambda$ is the largest eigenvalue of all $\hat{A}^\top(I - ee^\top)\hat{A}$ for each self-attention matrix $\hat{A}$, and $\sigma_1$, $\sigma_2$ are the minimum standard deviation for two layer normalization operations.*

*Proof.* From the definition of self-attention and feed-forward modules, we have

$$Attn(X) = \text{LayerNorm}(X + \sum_{k=1}^H \hat{A}^k XW^k + \mathbf{1}b^\top) = (X + \sum_{k=1}^H \hat{A}^k XW^k + \mathbf{1}b^\top - \mathbf{1}b_{LN}^\top)D_{LN}^{-1}$$

$$
\begin{aligned}
FF(X) &= \text{LayerNorm}(X + \text{ReLU}(XW_1 + \mathbf{1}b_1^\top)W_2 + \mathbf{1}b_2^\top) \\
&= (X + \text{ReLU}(XW_1 + \mathbf{1}b_1^\top)W_2 + \mathbf{1}b_2^\top - \mathbf{1}b_{LN}^\top)D_{LN}^{-1}
\end{aligned}
$$

Based on the Lemma 1, we have

$$d_{\mathcal{M}}(Attn(\boldsymbol{X})) = d_{\mathcal{M}}((\boldsymbol{X} + \sum_{k=1}^{h} \hat{\boldsymbol{A}}^k \boldsymbol{X} \boldsymbol{W}^k + \mathbf{1}\boldsymbol{b}^\top - \mathbf{1}\boldsymbol{b}_{LN}^\top)\boldsymbol{D}_{LN}^{-1})$$

$$\leq d_{\mathcal{M}}(\boldsymbol{X}\boldsymbol{D}_{LN}^{-1}) + d_{\mathcal{M}}(\sum_{k=1}^{h} \hat{\boldsymbol{A}}^k \boldsymbol{X} \boldsymbol{W}^k \boldsymbol{D}_{LN}^{-1}) + d_{\mathcal{M}}(\mathbf{1}(\boldsymbol{b} - \boldsymbol{b}_{LN})^\top)$$

$$\leq \sigma_1^{-1} d_{\mathcal{M}}(\boldsymbol{X}) + \sum_{k=1}^{h} d_{\mathcal{M}}(\hat{\boldsymbol{A}}^k \boldsymbol{X} \boldsymbol{W}^k \boldsymbol{D}_{LN}^{-1})$$

$$\leq \sigma_1^{-1} d_{\mathcal{M}}(\boldsymbol{X}) + \sqrt{\lambda} h s \sigma_1^{-1} d_{\mathcal{M}}(\boldsymbol{X})$$

$$= (1 + \sqrt{\lambda} h s) \sigma_1^{-1} d_{\mathcal{M}}(\boldsymbol{X}).$$

$$d_{\mathcal{M}}(FF(\boldsymbol{X})) = d_{\mathcal{M}}((\boldsymbol{X} + \mathrm{ReLU}(\boldsymbol{X}\boldsymbol{W}_1 + \mathbf{1}\boldsymbol{b}_1^\top)\boldsymbol{W}_2 + \mathbf{1}\boldsymbol{b}_2^\top - \mathbf{1}\boldsymbol{b}_{LN}^\top)\boldsymbol{D}_{LN}^{-1})$$

$$\leq d_{\mathcal{M}}(\boldsymbol{X}\boldsymbol{D}_{LN}^{-1}) + d_{\mathcal{M}}(\mathrm{ReLU}(\boldsymbol{X}\boldsymbol{W}_1 + \mathbf{1}\boldsymbol{b}_1^\top)\boldsymbol{W}_2\boldsymbol{D}_{LN}^{-1}) + d_{\mathcal{M}}(\mathbf{1}(\boldsymbol{b}_2^\top - \boldsymbol{b}_{LN}^\top)\boldsymbol{D}_{LN}^{-1})$$

$$\leq d_{\mathcal{M}}(\boldsymbol{X}\boldsymbol{D}_{LN}^{-1}) + d_{\mathcal{M}}(\boldsymbol{X}\boldsymbol{W}_1\boldsymbol{W}_2\boldsymbol{D}_{LN}^{-1}) + d_{\mathcal{M}}(\mathbf{1}\boldsymbol{b}_1^\top \boldsymbol{W}_2\boldsymbol{D}_{LN}^{-1})$$

$$\leq \sigma_2^{-1} d_{\mathcal{M}}(\boldsymbol{X}) + s^2 \sigma_2^{-1} d_{\mathcal{M}}(\boldsymbol{X})$$

$$= (1 + s^2) \sigma_2^{-1} d_{\mathcal{M}}(\boldsymbol{X}).$$

It follows that

$$d_{\mathcal{M}}(\boldsymbol{H}_{l+1}) \leq (1 + s^2)(1 + \sqrt{\lambda} h s) \sigma_1^{-1} \sigma_2^{-1} d_{\mathcal{M}}(\boldsymbol{H}_l).$$

$\square$

## B    DATA SET

### B.1    MNLI

The Multi-Genre Natural Language Inference (Williams et al., 2018) is a crowdsourced ternary classification task. Given a premise sentence and a hypothesis sentence, the target is to predict whether the last sentence is an [entailment], [contradiction], or [neutral] relationships with respect to the first one.

### B.2    QQP

The Quora Question Pairs (Chen et al., 2018) is a binary classification task. Given two questions on Quora, the target is to determine whether these two asked questions are semantically equivalent or not.

### B.3    QNLI

The Question Natural Language Inference (Wang et al., 2018b) is a binary classification task derived from the Stanford Question Answering Dataset (Rajpurkar et al., 2016). Given sentence pairs (question, sentence), the target is to predict whether the last sentence contains the correct answer to the question.

### B.4    SST-2

The Stanford Sentiment Treebank (Socher et al., 2013) is a binary sentiment classification task for a single sentence. All sentences are extracted from movie reviews with human annotations of their sentiment.

### B.5    COLA

The Corpus of Linguistic Acceptability (Warstadt et al., 2019) is a binary classification task consisting of English acceptability judgments extracted from books and journal articles. Given a single sentence, the target is to determine whether the sentence is linguistically acceptable or not.

### B.6 STS-B

The Semantic Textual Similarity Benchmark (Cer et al., 2017) is a regression task for predicting the similarity score (from 1 to 5) between a given sentence pair, whose sentence pairs are drawn from news headlines and other sources.

### B.7 MRPC

The Microsoft Research Paraphrase Corpus (Dolan & Brockett, 2005) is a binary classification task. Given a sentence pair extracted from online news sources, the target is to determine whether the sentences in the pair are semantically equivalent.

### B.8 RTE

The Recognizing Textual Entailment (Bentivogli et al., 2009) is a binary entailment classification task similar to MNLI, where [neutral] and [contradiction] relationships are classified into [not entailment].

### B.9 SWAG

The Situations with Adversarial Generations (Zellers et al., 2018) is a multiple-choice task consisting of 113K questions about grounded situations. Given a source sentence, the task is to select the most possible one among four choices for sentence continuity.

### B.10 SQuAD v1.1

The Stanford Question Answering Dataset (SQuAD v1.1) (Rajpurkar et al., 2016) is a large-scale question and answer task consisting of 100K question and answer pairs from more than 500 articles. Given a passage and the question from Wikipedia, the goal is to determine the start and the end token of the answer text.

### B.11 SQuAD v2.0

The SQuAD v2.0 task (Rajpurkar et al., 2018) is the extension of above SQuAD v1.1, which contains the 100K questions in SQuAD v1.1 and 50K unanswerable questions. The existence of unanswerable question makes this task more realistic and challenging.

## C IMPLEMENTATION DETAILS

The hyper-parameters of various downstream tasks are shown in Table 4.

Table 4: Hyper-parameters for different downstream tasks.

|  | GLUE | SWAG | SQuAD v1.1 | SQuAD v2.0 |
|---|---|---|---|---|
| Batch size | 32 | 16 | 32 | 48 |
| Weight decay | [0.1, 0.01] | [0.1, 0.01] | [0.1, 0.01] | [0.1, 0.01] |
| Warmup proportion | 0.1 | 0.1 | 0.1 | 0.1 |
| Learning rate decay | linear | linear | linear | linear |
| Training Epochs | 3 | 3 | 3 | 2 |
| Learning rate | [2e-5, 1e-5, 1.5e-5, 3e-5, 4e-5, 5e-5] | | | |

