# OpenReview forum: "Revisiting Over-smoothing in BERT from the Perspective of Graph"
_ICLR.cc/2022/Conference — ICLR 2022 Spotlight_

### Official Review · Reviewer_M9jn · 2021-10-23

**Correctness:** 4
**Technical Novelty And Significance:** 3
**Empirical Novelty And Significance:** 4
**Recommendation:** 8
**Confidence:** 4

**Main Review:**

This paper tries to understand the over-smoothing phenomenon of Transformer-based models such as BERT. The analysis is from the perspective of viewing BERT and Transformer as graph neural networks. By analogy and analysis of graph neural networks, authors show some theoretical analysis and find that layer
normalization plays a key role in the over-smoothing issue. Specifically, if the standard deviation of layer normalization is sufficiently large, the output of Transformer stacks will converge to a specific low-rank subspace
and results in over-smoothing. Then to alleviate the over-smoothing problem, authors
consider hierarchical fusion strategies, which combine the representations from
different layers adaptively to make the output more diverse.

An accept has been recommended for this paper for the interestingness of the topic and the novel perspective of theoretical study of BERT by graph neural networks.  The over-smoothing study with layernorm via graph is useful to both BERT, optimization, and graph community. This will inspire some new directions for researchers and hence an accept is recommended.

Authors however need to address several concerns in the review.
1. If LayerNorm cause issue, Why does it help then? What do you suggest to replace it, e.g., will using prenorm by Di He et al resolve the issue?

2. The concat and max fusion has appeared in the graph paper that author cited. It's interesting to see this can help BERT as well and in that sense this is novel but authors should clearly indicate where this has appeared before as well.



**Summary Of The Paper:**

This paper tries to understand the over-smoothing phenomenon of Transformer-based models such as BERT. The analysis is from the perspective of viewing BERT and Transformer as graph neural networks. By analogy and analysis of graph neural networks, authors show some theoretical analysis and find that layer
normalization plays a key role in the over-smoothing issue. Specifically, if the standard deviation of layer normalization is sufficiently large, the output of Transformer stacks will converge to a specific low-rank subspace
and results in over-smoothing. Then to alleviate the over-smoothing problem, authors
consider hierarchical fusion strategies, which combine the representations from
different layers adaptively to make the output more diverse.

**Summary Of The Review:**

An accept has been recommended for this paper for the interestingness of the topic and the novel perspective of theoretical study of BERT by graph neural networks. This will inspire some new directions for researchers and hence an accept is recommended.

---

> ### Author Response · Authors · 2021-11-20
> **Reply to Reviewer M9jn**
>
> Thank you for your thoughtful review and valuable feedback. We address your concerns as follows.
>
> --------------------
>
> Q1. **"If LayerNorm cause issue, Why does it help then?"**
>
> Layer normalization is helpful for neural network optimization (e.g., stable training) [1]. However, in this paper, we discuss the over-smoothing issue, which is another topic.
>
>
>
>
> --------------------
>
>
>
>
> Q2. **"What do you suggest to replace it, e.g., will using prenorm by Di He et al resolve the issue?"**
>
> As discussed in the first paragraph of Section 6, pre-norm doesn’t outperform post-norm empirically (Table 2,3,4,5 in [2]). In our understanding, the reason may be the uncontrollable variance due to the following summation:
>
> $$x_l = x_0+f_0(x_0)+f_1(x_1)+\dots+f_{l-1}(x_{l-1})$$
>
>
> In this paper, we preserve the post-norm scheme and overcome the over-smoothing issue by combining representations from different layers.
>
>
>
>
> --------------------
>
>
> Q3.  **"The concat and max fusion has appeared in the graph paper that author cited. It's interesting to see this can help BERT as well and in that sense this is novel but authors should clearly indicate where this has appeared before as well"**
>
>
> We think concat and max fusions are basic operations and only cite that paper in Related Work section before. We have clearly indicated as suggested in the revised version (Section 6.1).
>
>
>
>
>
> --------------------
>
>
>
> [1] J. Ba, J. Kiros, and G. Hinton. Layer normalization. Preprint arXiv:1607.06450, 2016.
>
>
> [2] R. He, A. Ravula, B. Kanagal and J. Ainslie. Realformer: Transformer likes residual attention. In Findings of Annual Meeting of the Association for Computational Linguistics, 2021.

---

### Official Review · Reviewer_g9XE · 2021-11-02

**Correctness:** 4
**Technical Novelty And Significance:** 3
**Empirical Novelty And Significance:** 3
**Recommendation:** 8
**Confidence:** 3

**Main Review:**

•	The authors have provided sufficient deductions to establish the theory of token over-smooth in deeper transformer layers. I feel convinced that layer normalization could lead to the over-smoothing issue. One question, it is not obvious to me that how theoretically fusion strategy would help prevent token over-smoothing. Could authors discuss the question in terms of v in equation 8? How does the distribution of the layer normalization term look like in the fusion models (replicate figure 5)? A further question is whether the fusion strategies improved the over smoothing issue by reducing the layer normalization relatively.

•	The second question is about the contribution of self-attention block and feedforward block to token over-smoothing. The theory in section 5 considered both the self-attention block and the feedforward block, and the layer normalization of both blocks seem to be equally contributed to token representation smoothing. However, the shared attention test in figure 3 seems to suggest a dominant role of the self-attention block. I’m wondering how authors interpret this.

**Summary Of The Paper:**

The current manuscript explored the token over-smoothing problem in Bert transformer blocks. Empirically, the authors showed that the token over-smooth problem is increasingly severe in deeper layers of a Bert model. The token over-smoothing is potentially harmful to task performance. As the over-smoothing problems have been discussed in graph networks extensively, the authors established a connection between transformer blocks with graph networks and looked at the over-smoothing problem of transformer blocks analogously. The authors revealed that layer normalization in the transformer blocks contributed to the over-smoothing issue both theoretically and empirically. Moreover, the authors proposed to improve the over-smoothing issue by hierarchical fusion strategies, that is utilize token representation from earlier layers. Overall, the author applied a novel approach to shed light on an insufficiently investigated problem. The paper was well written.

**Summary Of The Review:**

Overall, I vote for accepting this paper. I like the idea of establishing an analogy between self-attention and the adjacency matrix of a graph model, and I like the theory that establishes the relationship between layer normalization and dimensionality reduction of token representation. My major concern is about the clarity of some parts of the paper and some additional improvement. I hope the authors could address my concerns in the rebuttal.

---

> ### Author Response · Authors · 2021-11-20
> **Reply to Reviewer g9XE**
>
> Thank you for your thoughtful review and valuable feedback. We address your concerns as follows.
>
> ---------
>
> Q1. **"it is not obvious to me that how theoretically fusion strategy would help prevent token over-smoothing"**
>
> As for the theoretical analysis of our fusion method, please refer to the general reply.
>
>
>
>
>
> ---------
>
>
> Q2. **"Could authors discuss the question in terms of v in equation 8?" & "How does the distribution of the layer normalization term look like in the fusion models (replicate figure 5)?"**
>
> Even though Theorem 2 still holds for two successive layers in our fusion model, the selection of the final output is different. In comparison, vanilla BERT takes representation from the last layer as the final output, while we allow our fusion model to select representations from both shallow layers and deep layers as the final output. Therefore, v cannot indicate the severity of the over-smoothing in the fusion model.
>
> As suggested, we also replicate figure 5 for Self-gate Fusion model and report the mean of $\sigma_1\sigma_2$ in the following table.
>
>
>
>
> |   |                   STS-B  | CoLA |  SQuAD |
> |:----|:----:|:----:|:----:|
> |BERT     |             1.53 |      0.41   |    0.89 |
> |BERT (self-gate)    |  1.33  |    0.39    |   0.82 |
>
> As can be seen, $\sigma_1\sigma_2$ is smaller in the fusion model.
>
>
> ---------
>
> Q3. **"A further question is whether the fusion strategies improved the over smoothing issue by reducing the layer normalization relatively"**
>
> As discussed in Q2, $\sigma_1\sigma_2$ is reduced empirically. However, since our fusion model combines representation from both shallow layers and deep layers, $\sigma_1\sigma_2$ cannot indicate the severity of the over-smoothing completely.
>
>
> ---------
>
>
>
>
>
>
>
> Q4. **"the shared attention test in figure 3 seems to suggest a dominant role of the self-attention block"**
>
>
> Figure 3 measures the correlation of attention matrices between two successive layers, which doesn’t look at the feed-forward block. It is our observation that the self-attention matrix is similar across Transformer blocks, which is the relationship between self-attention and graph and the motivation for theoretical analysis in Section 5.
>
>
>
>
>
> From Theorem 2, we claim that the product of  $\sigma_1$ (from self-attention block) and $\sigma_2$ (from feedforward block)  is important. To illustrate the domination of these two terms, we compare the mean of them in the following table.
>
>
>
> |                 |  STS-B  | CoLA | SQuAD |
> |:----|:----:|:----:|:----:|
> |$\sigma_1$ |      0.55  |    0.32   |    0.46 |
> |$\sigma_2$|      1.73    |  1.27   |    1.61 |
>
>
>
> As can be seen, $\sigma_2$ is dominant empirically.
>
>
>
>
> ---------
>
>
>
> Q5. **"My major concern is about the clarity of some parts of the paper and some additional improvement"**
>
>
> We have clarified the above parts more clearly in the revised version (first paragraph in Section 6).
>
> ---------
>
> [1] R. He, A. Ravula, B. Kanagal and J. Ainslie. Realformer: Transformer likes residual attention. In Findings of Annual Meeting of the Association for Computational Linguistics, 2021.

---

### Official Review · Reviewer_8SR3 · 2021-11-07

**Correctness:** 4
**Technical Novelty And Significance:** 3
**Empirical Novelty And Significance:** 3
**Recommendation:** 6
**Confidence:** 3

**Main Review:**

Pros:

a) The theoretical analysis on BERT is a good contribution. The analysis methods is not completely new but applicable to analysis on transformers.

Cons:

a) Experimental results shown in  Table 1 and Table 2  do not show any significant improvement to overcome oversmothing. Most of the proposed method only give a small improvement which let me ask the question whether proposed fusion method is good effective enough for oversmoothing. Can authors provide further experiments with different datasets or applications to demonstrate the effectiveness of the proposed method?

b) Lack of theoretical support for the proposed fusion method. Is it possible to theoretically demonstrate that the fusion method overcome oversmoothing?

**Summary Of The Paper:**

This paper discusses the oversmoothing in transformer models such as BERT. It provides a theoretical analysis on the existence of oversmoothing in transformers and proposes a hierarchical fusion method as a solution to oversmoothing.

**Summary Of The Review:**

The analysis of oversmoothing in transformer is a good contribution. The limited empirical evidence and lack strong improvement over oversmoothing by the proposed method are the main limitations of this paper.

---

> ### Author Response · Authors · 2021-11-20
> **Reply to Reviewer 8SR3**
>
> Thank you for your thoughtful review and valuable feedback. We address your concerns as follows.
>
> --------
>
> Q1. **"Table 1 and Table 2 do not show any significant improvement" & "Can authors provide further experiments… to demonstrate the effectiveness of the proposed method?"**
>
>
>
>
>
> Table 1 is not our method to overcome over-smoothing. The main point of table 1 is to show the connection between self-attention and the graph, motivating the theoretical analysis in Section 5.
> Compared with vanilla BERT and ALBERT, self-attention matrix sharing is also a by-product that not only achieves similar performance but also reduces the FLOPs and parameters.
>
>
>
>
>
> Table 2 is our results on GLUE datasets including several sub-tasks. Since Theorem 2 claims that the over-smoothing issue is dependent on $v$,  the significance of improvement is affected by the severity of over-smoothing in the sub-task. For instance, the improvement on QQP is marginal ($+0.2$), which is because the over-smoothing issue is not serious on QQP. Overall, our fusion method consistently improves the performance over the sub-tasks.
>
> As you suggested, we perform our methods on SWAG [1], a benchmark for grounded commonsense inference. Here we report the accuracy and the token-wise cosine similarity of the output.
>
>
>
>
> |  |              BERT  |    BERT (concat)  |  BERT (max)  |  BERT (self-gate) |
> |:----|:----:|:----:|:----:|:----:|
> |Accuracy  |   81.6%   |      82.0%      |     81.9%   |         82.1% |
> |CosSim    |    0.61      |   0.43     |       0.50      |       0.39              |
>
>
> As shown in the above table, the fusion approaches still perform better than the baseline by alleviating the over-smoothing problem.
>
>
>  --------
>
>
> Q2. **"Lack of theoretical support for the proposed fusion method."**
>
> Please refer to the general reply.
>
>
>
>
>
> --------
>
>
> [1] Zellers, R., Bisk, Y., Schwartz, R., and Choi, Y. SWAG: A Large-Scale Adversarial Dataset for Grounded Commonsense Inference. In Empirical Methods in Natural Language Processing, 2018.

---

> > ### Author Response · Authors · 2021-11-29
> > **Reply to Reviewer 8SR3 - Any further questions**
> >
> > We would like to thank you again for your detailed reviews. We have updated our draft and added replies to your two Cons with our latest experimental results.
> >
> > Given that your current score is 5, we would appreciate it if you could let us know if our responses have addressed your concerns and whether you still have any other questions on the current draft.
> >
> > We would be happy to do any follow-up discussion or address any additional comments.

---

> > ### Comment · Reviewer_8SR3 · 2021-11-29
> > **Response**
> >
> > Thank you for the rebuttal.
> > I think the theory is good so I increased the score.

---

> > > ### Author Response · Authors · 2021-11-29
> > > **Thanks for your endorsement**
> > >
> > > Thanks for your response and we appreciate your endorsement.

---

### Official Review · Reviewer_9nYH · 2021-11-09

**Correctness:** 3
**Technical Novelty And Significance:** 3
**Empirical Novelty And Significance:** 3
**Recommendation:** 6
**Confidence:** 3

**Main Review:**

Pros:
1. Addressing the over-smoothing problem in BERT is a promising direction, which will further the research on the study of transformers.
2. The theoretical analysis on the over-smoothing problem in BERT from the graph perspective is new and provides interesting findings.

Cons:
1. The authors claim that they propose three layer fusion methods which have been explored in previous methods in my understanding. The authors should carefully claim that the fusion methods are "proposed" in this paper. A sufficient survey should be conducted on layer fusion of BERT.
2. The over-smoothing problem is neither well theoretically analyzed nor empirically proved to be addressed via layer fusion methods.

Questions:
1. Is it possible to theoretically prove that layer fusion can alleviate the over-smoothing problem like Section 5?
2. Is it possible to empirically verify the layer fusion methods' ability to address the over-smoothing problem on more datasets. In fact, I notice that Fig. 6 has proved cosine similarity between BERT and BERT(self-gate) on three datasets. However, the paper should provide more experimental setting information. In my understanding, since "the last layer is replaced with the aggregated representation", should the cosine similarity of BERT(self-gate) at the 12th layer be lower than some of the layers before (since the last layer is the aggregated representation of all the 12 layers)?

**Summary Of The Paper:**

This paper explored the over-smoothing problem in BERT from the perspective of graph. In detail, empirical analysis first has been provided for demonstrating over-smoothing exist in BERT. Then, the authors theoretically prove the observation through the comparison between self-attention and graph. The authors claim that "layer normalization plays a key role in the over-smoothing issue, namely, if the standard deviation of layer normalization is sufficiently large, the output of Transformer stacks will converge to a specific low-rank subspace and results in over-smoothing". To alleviate the over-smoothing issue, layer fusion methods (which have been widely explored in previous studies) have been presented and verified to be effective in downstream tasks. However, the over-smoothing problem is neither well theoretically analyzed nor empirically proved to be addressed via layer fusion methods.

**Summary Of The Review:**

A marginally above the acceptance threshold is given. Overall, this paper addresses the over-smoothing problem in BERT. Theoretical analysis on the over-smoothing problem in BERT from the perspective of graph is new and provides interesting findings. However, proposed fusion methods are not that novel and not well theoretically and empirically proved to be able to address the over-smoothing problem.

---

> ### Author Response · Authors · 2021-11-20
> **Reply to Reviewer 9nYH**
>
> Thank you for your thoughtful review and valuable feedback. We address your concerns as follows.
>
> --------
> Q1. **"The authors should carefully claim that the fusion methods are proposed in this paper and a sufficient survey should be conducted on layer fusion of BERT" & "proposed fusion methods are not that novel"**
>
> Since the main contribution of our paper is the theoretical analysis of the over-smoothing in BERT, we consider three intuitive hierarchical fusion strategies to alleviate over-smoothing. To the best of our knowledge, while fusion strategy has been around for a long time, our paper is the first that applies fusion to alleviate the over-smoothing problem in BERT.
>
> As you suggested, we have modified the claim and discussed fusion strategies in the graph field in the revised version (Section 6.1).
>
> --------
>
> Q2. **"The over-smoothing is not empirically proved to be addressed via layer fusion methods" & "verify the layer fusion methods' ability on more datasets"**
>
>
> We have empirically proved that in Figure 6. As can be seen, compared with BERT, the similarity between tokens is reduced and the performance is improved (shown in Table 2) by the fusion strategy.
>
>
>
>
> As suggested, we perform our methods on SWAG [2], which is a task for grounded commonsense inference. The performance and the token-wise cosine similarity of the output are illustrated in the following.
>
> |      | BERT   |   BERT (concat)  |  BERT (max)  |  BERT (self-gate) |
> |  :----  |  :----: | :----: | :----: | :----: |
> | Accuracy |       81.6%      |    82.0%    |       81.9%      |      82.1%|
> | CosSim    |       0.61    |      0.43       |     0.50      |       0.39  |
>
>
> As can be seen,  all three fusion methods can improve the performance on
> SWAG and also reduce the similarity between tokens.
>
>
>
> --------
>
>
>
> Q3. **"should the cosine similarity of BERT (self-gate) at the 12th layer be lower than some of the layers before (since the last layer is the aggregated representation of all the 12 layers)?"**
>
>
> Yes. As discussed in Section 6.1, our final output is the combination of all 12 layers and we omit the 12th layer for fusion strategy in Figure 6. The table below shows the token-wise similarity at the 12th layer and at the final output.
>
>
>
> |Task   |        12th layer        |  final output          | $\Delta$|
> |  :----  |  :----: | :----: | :----: |
> |STS-B |          76.0%           |     63.7%        |     -12.3%|
> |CoLA|           47.6%           |    46.1%            | -1.5%|
> |SQuAD|          73.4%           |     71.0%           | -2.4% |
>
>
>
>
>
> As shown in the table,  the fusion strategy leads to a reduction over the token-wise similarity. The reduction is the most significant on STS-B as the over-smoothing problem is the most serious compared with other tasks (as shown in Figure 5).
>
>
>
>
> --------
>
>
>
> Q4. **"Layer fusion theoretically addresses over-smoothing."**
>
>
> Please refer to the general reply.
>
>
> --------
>
>
>
>
>
> Q5. **"the paper should provide more experimental setting information."**
>
> We have added experiment setting information discussed above (Q3) in the revised version (Section 6.2.2).
>
>
>
>
>
>
> --------
>
>
>
> [1] Z. Liu, Y. Lin, Y. Cao, H. Hu, Y. Wei, Z. Zhang, S. Lin and B. Guo. Swin Transformer: Hierarchical Vision Transformer using Shifted Windows. In International Conference on Computer Vision, 2021.
>
> [2] R. Zellers, Y. Bisk, R. Schwartz, and Y. Choi. SWAG: A Large-Scale Adversarial Dataset for Grounded Commonsense Inference. In Empirical Methods in Natural Language Processing, 2018.

---

> > ### Comment · Reviewer_9nYH · 2021-11-30
> > **Thanks for your response and efforts on revising the paper.**
> >
> > After reading your response to question 3, I recommend the authors add the 12th layer's similarity in Figure 6. From the additional provided results of the similarity of the 12th layer, we find that the fusion outputs are only less 'over-smoothing' than the 12th layer on STS-B and SQuAD, and the 11th and 12th layers on COLA.
> > Actually, the fusion strategy pays more attention to higher layers as shown in Figure 7. However, these layers are more likely to be over-smoothing. It indicates that higher layers' features are more ‘dominant for representation’ but ‘over-smoothing’. This inherent conflict makes it hard on the trade-off between the dominant layer selection and the degree of its 'over-smoothing.

---

> > > ### Author Response · Authors · 2021-11-30
> > > **Thanks for your suggestion**
> > >
> > > Agreed, we will add the 12th layer's similarity in Figure 6.
> > >
> > > Thanks for your response and we appreciate your endorsement.

---

### Author Response · Authors · 2021-11-20
**General Reply - Fusion strategy theoretically alleviates the over-smoothing issue**

Here we take Concat Fusion as an example and prove that our fusion strategy can alleviate the over-smoothing issue theoretically.

$\textbf{Claim}$. *Consider a $L$-layer Concat Fusion variant, there exists parameterizations for which CosSim($\sum_{i=1}^L\alpha_iH_i$)<CosSim($H_L$), which overcomes the over-smoothing issue.*

A simple example of such as parametrization can be [$\alpha_1$, $\alpha_2$, $\dots$, $\alpha_L$] = [1, 0, $\dots$, 0],
in which case CosSim($\sum_{i=1}^L\alpha_iH_i$)=CosSim($H_1$)<CosSim($H_L$).

Similarly, Self-gate Fusion can reduce the CosSim by selecting representations from prior layers. We learn the importance weights in an end-to-end manner. Empirically, the illustration of overcoming the over-smoothing issue is shown in Figure 6 and the importance weights are visualized in Figure 7.

---

### Public Comment · ~Kuangqi_Zhou1 · 2022-03-08
**Related work**

This is an interesting paper! Taking a graph perspective in studying over-smoothing of Transformer-based models is a novel idea.

We have a work showing that the variance-scaling operation in layer normalization is the key to alleviate the issue of GCNs. We believe our work is related to yours, and would like to share it with you. If you think it interesting, we hope you could include it in the references of the camera-ready version of your work. Thank you.

Understanding and Resolving Performance Degradation in Deep Graph Convolutional Networks, CIKM 2021. https://arxiv.org/pdf/2006.07107.pdf

---

### Decision · Program_Chairs · 2022-01-20

**Decision:**

Accept (Spotlight)

**Comment:**

This paper has a deep analysis of the over-smoothing phenomenon in BERT from the perspective of graph. Over-smoothing refers to token uniformity problem in BERT, different input patches mapping to similar latent representation in ViT and the problem of shallower representation better than deeper (overthinking). The authors build a relationship between Transformer blocks and graphs. Namely, self-attention matrix can be regarded as a normalized adjacency matrix of a weighted graph. They prove that if the standard deviation in layer normalization is sufficiently large, the outputs of the transformer stack will converge to a low-rank subspace, resulting in over-smoothing.

In this paper, they also provide theoretical proof why higher layers can lead to over-smoothing. Empirically , they investigate the effects of the magnitude of the two standard deviations between two consecutive layers on possible over-smoothing in diverse tasks.

In order to overcome over-smoothing, they propose a series of hierarchical fusion strategy that adaptively fuses presentation from different layers, including concatenation fusion, max fusion and self-gate fusion into post-normalization. These strategies reduce similarities between tokens and outperforms BERT baseline on a few datasets (GLUE, SWAG and SQuAD).

Overall I agree with reviewers that this is a good contribution.